# Effect of Surfactant on Bubble Formation on Superhydrophobic Surface in Quasi-Static Regime

**DOI:** 10.3390/biomimetics10060382

**Published:** 2025-06-07

**Authors:** Hangjian Ling, John Ready, Daniel O’Coin

**Affiliations:** Department of Mechanical Engineering, University of Massachusetts Dartmouth, Dartmouth, MA 02747, USA; jready@umassd.edu (J.R.); docoin@umassd.edu (D.O.)

**Keywords:** bubble formation, superhydrophobic surface, surfactant, surface tension, contact angle

## Abstract

We experimentally studied the effect of a surfactant on bubble formation on a superhydrophobic surface (SHS). The bubble was created by injecting gas through an orifice on the SHS at a constant flow rate in the quasi-static regime. The surfactant, 1-pentanol, was mixed with water at concentration *C* ranging from 0 to 0.08 mol/L, corresponding to surface tension *σ* ranging from 72 to 43 mN/m. We found that as *C* increased, the bubble detachment volume (*V*_d_) and maximum bubble base radius (*R*_d_^max^) decreased. For a low surfactant concentration, the static contact angle *θ*_0_ remained nearly constant, and *V*_d_ and *R*_d_^max^ decreased due to lower surface tensions, following the scaling laws *R*_d_^max^~*σ*^1/2^ and *V*_d_~*σ*^3/2^. The bubble shapes at different concentrations were self-similar. The bubble height, bubble base radius, radius at the bubble apex, and neck radius all scaled with the capillary length. For high surfactant concentrations, however, *θ*_0_ was greatly reduced, and *V*_d_ and *R*_d_^max^ decreased due to the combined effects of reduced *θ*_0_ and smaller *σ*. Lastly, we found that the surfactant had a negligible impact on the forces acting on the bubble, except for reducing their magnitudes, and had little effect on the dynamics of bubble pinch-off, except for reducing the time and length scales. Overall, our results provide a better understanding of bubble formation on complex surfaces in complex liquids.

## 1. Introduction

Recently, superhydrophobic surfaces (SHSs), inspired by lotus leaves and animal skins, have been proven to be a new approach to control bubble formation [1,2,3,4,5] and bubble movement [6]. An SHS exhibits a large static contact angle of *θ*_0_ > 150° due to a combination of surface roughness and hydrophobic chemistry [7]. Due to the large *θ*_0_, the SHS facilitates the formation of larger bubbles than do hydrophilic and hydrophobic surfaces. SHSs have found a wide range of applications, from reducing drag in turbulent flows [8,9,10,11] to enhancing boiling heat transfer [1,2] and preventing engineered surfaces from undergoing biofouling [12] and icing [13]. Most of these applications take advantage of the presence of the gas layer on SHSs. Understanding bubble formation on SHSs could potentially provide new solutions to sustain the gas layer on the surface and thus benefit the broader applications of SHSs. Furthermore, controlling bubble formation in liquid solutions is crucial for many other industrial and biomedical applications, including pool boiling heat transfer [14], surface cleaning [15], chemical processing [16], mineral particle recovery [17], and drug delivery [18].

Bubble formation due to gas injection through a nozzle or orifice has been extensively investigated [19,20]. The detached bubble volume (*V*_d_) is affected by many parameters, including the liquid properties [21,22,23,24,25], surface tension [26], gravity [27], surface wettability [28,29,30], orifice configuration [31,32,33], and gas flow rate [34,35]. Depending on the gas flow rate *Q*, the bubble formation can be separated into two regimes [36]: the quasi-static regime for small *Q* and the dynamic regime for large *Q*. In the quasi-static regime, the main forces acting on the bubble are due to surface tension and hydrostatic pressure, and *V*_d_ is independent of the gas flow rate [36]. In the dynamic regime, the forces due to the momentums of liquid and gas cannot be ignored, and *V*_d_ increases with the gas flow rate [37]. Depending on the surface wettability, bubble formation follows two modes [38]: the pinning mode for well-wetting surfaces (i.e., hydrophilic) and the de-pinning mode for poorly wetting surfaces (i.e., hydrophobic). In the pinning mode, the bubble base remains pinned at the rim of the orifice, whereas in the de-pinning mode, the bubble base expands beyond the orifice rim, leading to a larger *V*_d_ [39,40]. Furthermore, for hydrophobic surfaces, with increasing static contact angle *θ*_0_, the bubble base radius increases, resulting in a larger *V*_d_ [38].

Bubble formation on SHSs has received growing interest recently. A few researchers [41,42,43,44] investigated the effect of the size of the SHS region on bubble formation under a quasi-static regime. They found that with increasing size of the SHS, the bubble formation transitions from the pinning mode, where the bubble base is pinned at the rim of the SHS, to the de-pinning mode, where the bubble base moves along the SHS. Breveleri et al. [45] studied the bubble formation on an SHS fabricated on a porous material and found an increase in the bubble size with increasing pressure difference across the porous surface. Recently, O’Coin and Ling [46] examined the effect of the gas flow rate on bubble formation on an SHS under the quasi-static regime, and they found an increase in *V*_d_ with increasing gas flow rate.

However, the effect of surfactants on bubble formation on SHSs has not been studied, despite the fact that surfactants are widely present in natural environments and are used in many engineering systems. Surfactants typically adsorb at the gas–liquid interface due to their amphiphilic properties and are well-known for reducing the surface tension. The effect of surfactants on bubble formation at the nozzle and orifice has been extensively studied [47,48,49,50,51,52,53,54,55]. It is generally agreed that at low gas flow rates, a small amount of surfactant greatly reduces the detached bubble size due to the reduction of surface tension [52,54]. However, at high gas flow rates, due to the reduced time scale of bubble formation, the surfactants do not have sufficient time to reach the equilibrium concentration at the bubble surface. As a result, at high gas flow rates, the detached bubble size in surfactant solution is close to that in pure water [52]. In addition to reducing surface tension, surfactants have been shown to reduce the static contact angle of SHSs [56,57]. Therefore, surfactants may significantly alter bubble formation on SHSs due to the changes in the surface tension and static contact angle.

The goal of this experimental study is to investigate, for the first time, the effect of surfactants on bubble formation on SHSs. We use 1-pentanol as the surfactant model and focus on bubble formation at a low gas flow rate. The time scale of bubble formation is much longer than the time needed for the surfactant to reach the equilibrium concentration, rendering the effect of dynamic surface tension negligible. We apply high-speed imaging to capture the bubble shape, measure the bubble geometrical parameters by processing the recorded images, and calculate the forces acting on the bubble. We show that for low surfactant concentrations, the static contact angle *θ*_0_ remains nearly constant, the bubble shapes at different concentrations are self-similar, and the bubble size scales with the capillary length. However, for high surfactant concentrations, *θ*_0_ is greatly reduced, and the bubble size decreases due to the combination effect of smaller *θ*_0_ and smaller *σ*. We also discuss the effect of 1-pentanol on the dynamics of bubble pinch-off and the forces acting on the bubble.

## 2. Materials and Methods

The experimental setup for measuring bubble formation in the surfactant solution is shown in Figure 1a. This setup was also employed in our previous studies [44,46]. A transparent acrylic tank with inner dimensions of 100 mm by 100 mm was filled with the surfactant solution (density *ρ*_L_ = 997 kg/m^3^) to a height of 70 mm. A superhydrophobic surface (SHS) with dimensions of 50 mm by 50 mm was installed at the bottom of the tank. The SHS was fabricated by sandblasting an aluminum surface with an abrasive medium of grit size 60 (particle mesh size 35 to 100), followed by spray-coating the textured surface with hydrophobic nanoparticles (SOFT99 Corp, La Habra, CA, USA, Glaco Mirror Coat Zero). The aluminum surface was cleaned in an ultrasonic bath before and after the sandblasting. Figure 1b,c show the surface texture of the SHS obtained by scanning electron microscopy. Clearly, the SHS exhibited a combination of micro- and nano-scale surface roughness. When submerged in water, the SHS maintained a Cassie–Baxter state and trapped a thin layer of air. The experiments were performed at room temperature of *T* = 25 °C.

The surfactant solution was prepared by mixing 1-pentanol (Sigma-Aldrich, St. Louis, MO, USA, #398268, >99%, CAS #71-41-0) with deionized (DI) water. We selected 1-pentanol as the surfactant model due to its well-characterized kinetics of dynamic adsorption [58,59]. The concentration of 1-pentanol in the solution, denoted by *C*, ranged from 0 to 0.08 mol/L, where *C* = 0 corresponded to pure water. Due to the low concentration used, the addition of surfactant to water caused no significant changes in physical properties, except for the surface tension. The equilibrium surface tension between the air and the 1-pentanol solution, denoted by *σ*, was measured using the pendant droplet method [60]. The pendant drop method was calibrated by measuring the surface tension of pure water. For each concentration, the measurement was repeated at least five times, and the uncertainty was less than 2 mN/m. The surface tension for pure water was found to be 72 mN/m, which agreed well with the typically reported value. Figure 2a shows the measured *σ* as a function of *C*. The results showed good agreement with the Szyszkowski model [61], expressed as*σ* = *σ*_0_ − ℛ*T*/*ω* ln(1 + *kC*)(1)
where *σ*_0_ denotes the surface tension of pure water, ℛ is the ideal gas constant, *T* is the temperature, *ω* is the limiting partial molar area of the solute at the surface, and *k* is the solute adsorption coefficient. For 1-pentanol [61], *ω* = 1.48 × 10^5^ m^2^/mol and *k* = 66 L/mol. Furthermore, we estimated the time scale *τ*_e_ for reaching the equilibrium interfacial surfactant concentration based on the following model [62]:*τ*_e_ ~ (*k_a_C*/*Γ*_∞_ + *k_d_*)^−1^(2)
where *k_a_* is the adsorption kinematic coefficient, *k_d_* is the desorption kinematic coefficient, and *Γ*_∞_ is the maximum interfacial surfactant concentration. For 1-pentanol solutions with different concentrations, *k_a_*, *k_d_*, and *Γ*_∞_ are constants [58,59]: *k_a_* = 3.00 × 10^−3^ cm/s, *k_d_* = 1.10 × 10^2^ s^−1^, and *Γ*_∞_ = 5.90 × 10^−10^ mol/cm^2^. According to this model, for the current range concentration, *τ*_e_ < 0.006 s, which is more than two orders of magnitude smaller than the time scale of the bubbling period (~1 s). Therefore, we can safely assume that the surfactant concentration at the bubble surface reaches equilibrium during the bubble formation.

Figure 2b shows the static contact angle *θ*_0_ of the SHS as a function of *C*. The static contact angle was measured by the sessile-drop-fitting method with a small droplet (~5 μL) seated on the surface [63]. With increasing *C* from 0 to 0.05 mol/L, there was only a slight decrease in the static contact angle from 159 ± 2° to 155 ± 2°. However, with a further increase in the concentration to *C* = 0.08 mol/L, *θ*_0_ decreased significantly to 131 ± 2°, consistent with the observation by Mohammadi et al. [56]. This decrease in the static contact angle is likely due to the adsorption of surfactant on the SHS and the resulting reduction in interfacial energy between the liquid and solid. Previous studies [38] have also shown that for bubble formation on a hydrophobic surface (*θ*_0_ > 90°), the detached bubble volume increases with increasing *θ*_0_. Therefore, as will be shown later, for low concentrations *C* < 0.05 mol/L, where the change in *θ*_0_ is small, the detached bubble volume is mainly affected due to the change in surface tension. However, at high concentrations *C* = 0.08 mol/L, since *θ*_0_ is greatly reduced, the detached bubble volume decreases because of a combined effect of reduced surface tension and reduced *θ*_0_.

The bubble was created on the SHS by injecting air (density *ρ*_G_ = 1.2 kg/m^3^) through a 0.5 mm diameter orifice located at the center of the SHS at a constant gas flow rate. The size of the orifice was consistent with that used in the literature by Rubio-Rubio et al. [41] and Qiao et al. [43], who also investigated bubble formation on superhydrophobic surfaces. We expect that for superhydrophobic surfaces with large static contact angles (>150°), the orifice size has a negligible impact on the detached bubble size. However, for small static contact angles (<150°), the orifice size may influence bubble formation as shown by Heidarian et al. [28]. The effect of the orifice size on bubble formation is left for future investigation. A syringe pump (Model #NE-1010 SyringeONE, by New Era Pump System Inc., Farmingdale, NY, USA) was used to supply gas. To ensure a constant flow rate, the air passed through a long needle (152 mm in length, inner diameter 0.61 mm) before reaching the orifice. The bubble formation was recorded by a high-speed camera (PCO.dimax S4, Howell, MI, USA, pixel size 11 µm, 2016 × 2016 pixels) at a frame rate of 1000 frames per second. The spatial resolution of the imaging system was 34 µm/pixel. A collimated light (Thorlabs, model #QTH10, power 50 mW) was used for illumination. The data was recorded after a series of bubbles had formed and detached from the surface. The light was switched on briefly and did not change the temperature of the solution. The size of the bubble was much smaller than that of the tank. We expected that the side walls of the tank would have a negligible impact on bubble formation. The bubble formation was in the single-bubbling regime, and previously formed bubbles had minimal influence on the observed one. All experiments were performed at room temperature of 25 ± 1 °C. The acrylic tank was rinsed with deionized water at least three times prior to the experiments to ensure the purity of the surfactant solution.

Table 1 summarizes the main experimental parameters, including the 1-pentanol concentration (*C*), surface tension (*σ*), and static contact angle (*θ*_0_). The key experimental results, including the maximum bubble base radius (*R*_b_^max^) and bubble detached volume (*V*_d_), are also provided in the table. *R*_b_^max^ can be analogized to the nozzle radius for a bubble detached from a nozzle. We calculated the critical gas flow rate for the transition from the quasi-static regime to the dynamic regime as follows [36]: *Q*_cr_ = *π* (16/3g^2^)^1/6^ (*σ**R*_b_^max^/*ρ*_L_)^5/6^, where g = 9.78 m^2^/s is the gravitational acceleration. In the current study, the gas flow rate was *Q* < 30 mL/min, corresponding to *Q*/*Q*_cr_ < 0.07. The Weber number *We*=*ρ*_L_*Q*^2^/*π**σ**R*_b_^max3^ (ratio of gas momentum to surface tension) did not exceed 2 × 10^−3^. At this small flow rate, the bubble formation was in the quasi-static regime, and *Q* had little influence on the detached bubble volume [46]. The capillary length scale, defined as *l*_*σ*_ = (*σ*/*ρ*_L_*g*)^0.5^, is also provided in Table 1.

## 3. Results and Discussion

Figure 3 shows the time evolution of bubble shapes for five different 1-pentanol concentrations: *C* = 0, 0.01, 0.02, 0.05, and 0.08 mol/L. With increasing *C*, the bubble size, along with the detached volume, gradually decreased. For all cases, the bubble formation progressed through two stages: an initial expansion stage and a final necking stage. During the expansion stage (0 < *t* < 0.99*T*_0_), the bubble grew in both the horizontal and vertical directions due to the gas injection. The bubble was in the quasi-static condition due to the balance between the surface tension and hydrostatic pressure. However, during the necking stage (0.99*T*_0_ < *t* < *T*_0_), the bubble reached its maximum equilibrium volume and could no longer maintain the quasi-static condition. Instead, the bubble shape changed dramatically under the impact of surface tension and buoyancy. The necking process can be characterized by the formation of a neck near the bubble base and a rapid reduction in the neck radius. The necking stage occupied a very small portion of the whole bubbling period. In all cases, due to the constant gas flow rate, the bubble volume increased linearly with time.

Figure 4a shows the maximum bubble base radius (*R*_b_^max^) as a function of *C*. Clearly, with increasing *C*, *R*_b_^max^ reduced due to the reduced surface tension. A similar trend was observed for a hydrophobic surface made of Teflon by Mirsandi et al. [29]. Figure 4b shows the bubble detached volume (*V*_d_) as a function of *C*. For comparison, the Tate volume [64] *V*_T_ = 2*π**R*_b_^max^*σ*/(*ρ*_L_ − *ρ*_G_)g, derived based on the balance between the surface tension force and buoyancy force, was also plotted. Note that the effect of the contact angle in the expression of the Tate volume was ignored. Clearly, the trend that *V*_d_ decreased with increasing *C* was captured by *V*_T_. This agreement suggests that a reduction in *V*_d_ results from two factors: the decrease in *σ* and the decrease in *R*_b_^max^.

Figure 4c and d show *R*_b_^max^/*l*_*σ*_ and *V*_d_/*l*_*σ*_^3^ as a function of *C*, respectively. Interestingly, for low concentrations *C* < 0.05 mol/L, *R*_b_^max^/*l*_*σ*_ and *V*_d_/*l*_*σ*_^3^ were nearly constant, suggesting that the capillary length was a proper length scale for normalization. Therefore, we derived the scaling laws for *R*_b_^max^ and *V*_d_ in terms of surface tension as follows:*R*_b_^max^ ~ *l*_*σ*_ ~ *σ*^1/2^(3)*V*_d_ ~ *l*_*σ*_^3^ ~ *σ*^3/2^(4)

To confirm the proposed scaling relations, Figure 5a and b present *R*_b_^max^ and *V*_d_ as a function of *σ*, respectively. Clearly, for *C* < 0.05 mol/L, the two scaling laws had good agreement with the measurements. However, for the highest concentration *C* = 0.08 mol/L, both *R*_b_^max^ and *V*_d_ were much smaller than the prediction based on the scaling laws. This is attributed to the fact that, at the highest concentration, the static contact angle was significantly reduced, leading to a further decrease in *R*_b_^max^ and *V*_d_. In general, the detached bubble size depends on both the static contact angle [38] and surface tension. We expect that the scaling laws shown in Equations (3) and (4) hold for low surfactant concentrations, where the static contact angle is not significantly modified and only the surface tension is reduced due to the presence of surfactants.

Figure 6 shows the evolutions of the bubble geometrical parameters, including the bubble height *H*, bubble base radius *R*_b_, and radius at the bubble apex *R*_a_, as a function of bubble volume *V* for four different 1-pentanol concentrations: *C* = 0, 0.02, 0.05, and 0.08 mol/L. In Figure 6a–c, results are shown in the dimensional form. In Figure 6d–f, results are shown in the dimensionless form, where *l*_*σ*_ was chosen as the characterized length scale for normalization. Since the volume was linearly proportional to time, the curves had the same trends when plotted as a function of time. To measure these geometrical parameters, we employed an image processing method described in our previous work [44]. The procedure mainly involved a binarization of the image based on an intensity threshold to segment the region belonging to the bubble. *V* was calculated by accumulating the cross-sectional area at each height level from the bottom to the top of the bubble. *R_a_* was determined by fitting the bubble apex with a circle of radius *R*_a_. The uncertainty for determining the boundary of the bubble was less than 1 pixel or 34 µm.

As shown in Figure 6, the curves had similar trends regardless of the value of *C*. During the bubble growing period, with increasing volume, *H* increased linearly with *V*, *R*_a_ gradually decreased, and *R*_b_ experienced an initial increase due to bubble expansion along the horizontal surface and then became nearly constant. During the necking period, with increasing volume, all geometrical parameters changed dramatically, including a sudden increase in *H* and *R*_a_ and a sudden decrease in *R*_b_ due to the retraction of the contact line along the surface. With increasing *C*, all the geometrical parameters decreased due to the reductions in the surface tension and static contact angle. Interestingly, as shown in Figure 6d–f, showing the results in the dimensionless form, for low 1-pentanol concentrations *C* < 0.05 mol/L, the curves collapsed. This result implies that, for low concentrations *C* < 0.05 mol/L, the bubble shapes were self-similar. The capillary length was an appropriate scale for normalization, and the surfactant did not affect the kinematics of bubble formation, except for reducing the length scale. For the highest concentration *C* = 0.08 mol/L, the normalized curves did not overlap with others. This deviation is likely due to the significantly lower static contact angle compared to the other cases.

Figure 7 shows the evolution of the contact angle (*θ*_b_) at the bubble base as a function of bubble volume for four different 1-pentanol concentrations. *θ*_b_ was measured by applying second-order polynomial fitting to the bubble shape near the contact point with the solid surface. Clearly, for *C* < 0.05 mol/L, as the bubble volume increased and as the bubble base expanded along the horizontal surface, *θ*_b_ was nearly a constant, consistent with the observation by Rubio-Rubio et al. [41]. Furthermore, with increasing *C*, the trend in *θ*_b_ was consistent with the trend in *θ*_0_. For low 1-pentanol concentrations *C* < 0.05 mol/L, the effect of *C* on *θ*_b_ was small. For the highest 1-pentanol concentration *C* = 0.08 mol/L, *θ*_b_ was much smaller than it was in other cases.

Figure 8a shows the evolution of the bubble shape just before detachment (or pinch-off) for *C* = 0. The time to bubble pinch-off was defined as *τ*, with *τ* = 0 as the time of pinch-off. A neck formed near the base of the bubble at *τ*~20 ms. Approaching the pinch-off, the neck radius decreased at an accelerated rate. We measured the minimum neck radius and defined it as *R*_n_. Figure 8b shows the time evolutions of *R*_n_ for four different values of *C*. For all cases, as expected, *R*_n_ decreased to 0 as *τ* approached 0. Moreover, with increasing *C*, in agreement with the smaller bubble size, *R*_n_ decreased. However, as shown in Figure 8c, after the normalization of *R*_n_ by *R*_b_^max^ and *τ* by the capillary–inertia time scale *τ*_n_ = (*ρ*_L_*R*_b_^max 3^/*σ*)^0.5^, all the profiles collapsed nicely and roughly agreed with the trends observed for the pinch-off of a bubble from a nozzle [65]. All profiles exhibited a power-law relation, *R*_n_ ~ *τ*_n_^0.57^, where the exponent 0.57 is consistent with values reported in the literature [65,66,67,68] for the pinch-off of a bubble from a nozzle. This result suggests that the necking process was primarily driven by liquid inertia [66] and that the surface tension had a negligible impact on the dynamics of necking, except for reducing the length and time scales.

Finally, following our previous work [44], we performed a force balance analysis for the bubble. In the current study, since the gas flow rate was low and the bubble formation was in the quasi-static regime, the dynamic forces due to the momentums and viscosities of gas and liquid were negligible. We mainly calculated three static forces acting on the bubble: (1) the pressure force *F*_P_ = (2*σ*/*R*_a_ + *ρ*_G_g*H*)*π**R*_b_^2^ mainly caused by the surface tension at the bubble apex; (2) the surface tension force *F*_S_*=* −2*π**R*_b_*σ*sin*θ*_b_ applied at the rim of the bubble base; and (3) the buoyancy force *F*_B_ = (*ρ*_L_ − *ρ*_G_)g*V* − *ρ*_L_g*H**π**R*_b_^2^ due to the imbalance in the hydrostatic pressure applied on the bubble surface. Note that *F*_P_ is in the upward direction since the pressure in the bubble is higher than that in the liquid due to surface tension, and *F*_B_ is in the downward direction since the hydrostatic pressure applied on the bubble is in the downward direction.

Figure 9a and b show the three forces (*F*_P_, *F*_S_, and *F*_B_) as well as their summation *F*_P_ + *F*_B_ + *F*_S_ as a function of bubble volume for two 1-pentanol concentrations: *C* = 0 and 0.08 mol/L. The forces were normalized by the buoyancy force (*ρ*_L_ − *ρ*_G_)g*V*_d_, which corresponded to the detached bubble volume. Clearly, regardless of the value of *C*, the growth of the bubble was governed by similar forces: one lifting force *F*_P_ and two retaining forces *F*_S_ and *F*_B_. The three forces were roughly in balance (*F*_P_ + *F*_B_ + *F*_S_ was close to 0). As the bubble volume increased, the magnitudes of *F*_P_ and *F*_B_ increased linearly with the volume, while *F*_S_ quickly reached a nearly constant value. This result confirms that the dynamic forces due to the momentums of the injected gas and the surrounding liquid were negligible. Furthermore, this result suggests that the surface tension and static contact angle had little impact on the dynamics of bubble formation, except for reducing the magnitude of the forces.

## 4. Conclusions

In summary, we experimentally investigated the effect of 1-pentanol on bubble formation on a superhydrophobic surface (SHS) in the quasi-static regime. To quantify the bubble formation, we measured various geometrical parameters, including the bubble height, bubble base radius, radius at the bubble apex, contact angle, neck radius, and detached volume. We also calculated the forces acting on the bubble based on these geometrical parameters. Our key findings are summarized below:As the surfactant concentration increased, all bubble geometrical parameters, including the bubble detached size, decreased.For low surfactant concentrations, *θ*_0_ remained nearly constant, bubble shapes at different concentrations were self-similar, and *R*_d_^max^ and *V*_d_ decreased following the scaling laws *R*_d_^max^~*σ*^1/2^ and *V*_d_~*σ*^3/2^, respectively.For high surfactant concentrations, *θ*_0_ was greatly reduced, and *R*_d_^max^ and *V*_d_ decreased due to the combined effects of reduced surface tension and a lower static contact angle.The surfactant had little impact on the dynamics of bubble pinch-off, except for reducing the time and length scales. The minimum neck radius followed a power-law relation, *R*_n_~*τ*^0.57^, similar to that observed for bubble pinch-off from a nozzle.The surfactant had negligible impact on the forces acting on the bubble, except for reducing their magnitudes. The growth of the bubble was governed by a balance between surface tension and hydrostatic pressure.

Overall, our results provide deeper insights into bubble formation on complex surfaces in complex solutions. However, our study was limited to one type of surfactant (nonionic 1-pentanol), one type of SHS, and the quasi-static regime. Future studies should investigate the effects of surfactant types (anionic, cationic, and nonionic), textural geometries of the SHS, and gas flow rates on bubble formation. Regardless of the surfactant type or surface treatment, we can conclude from our results that the surfactant affects bubble formation through two main mechanisms: (1) reducing surface tension due to surfactant adsorption at the gas–liquid interface and (2) reducing the static contact angle due to surfactant adsorption at the solid–liquid interface. Both effects depend on the surfactant concentration and lead to a reduction in the detached bubble size. The scaling law shown in Figure 5 holds at low surfactant concentrations, where the static contact angle is not significantly modified. Furthermore, we expect that decreasing the solid fraction of the SHS, which increases the static contact angle (according to the Cassie–Baxter model), leads to an increase in the detached bubble size. As the gas flow rate increases to the dynamic regime, the effect of the surfactant on bubble formation may weaken, as the surfactant may not have sufficient time to reach the equilibrium concentration at the gas–liquid interface.

## Figures and Tables

**Figure 1 biomimetics-10-00382-f001:**
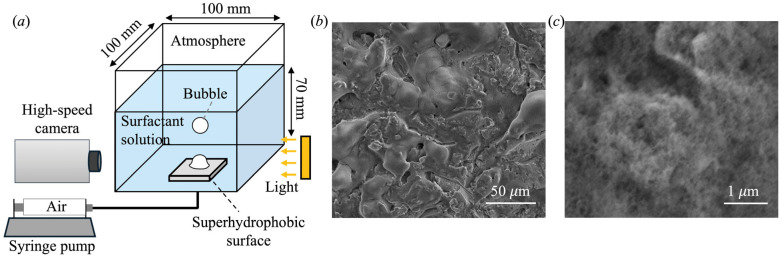
(**a**) Schematic drawing of experimental setup for measuring bubble formation on superhydrophobic surface; (**b**,**c**) SEM images of superhydrophobic surface.

**Figure 2 biomimetics-10-00382-f002:**
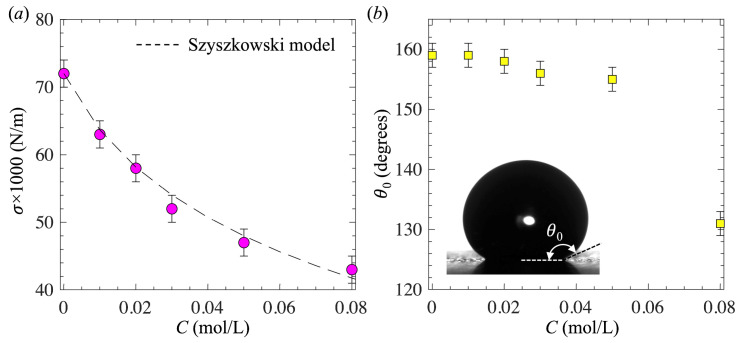
(**a**) Surface tension as a function of the 1-pentanol concentration and (**b**) the static contact angle of the superhydrophobic surface as a function of the 1-pentanol concentration. The dashed line in (**a**) is the Szyszkowski model [61]. The inserted image in (**b**) is a water droplet seated on the SHS.

**Figure 3 biomimetics-10-00382-f003:**
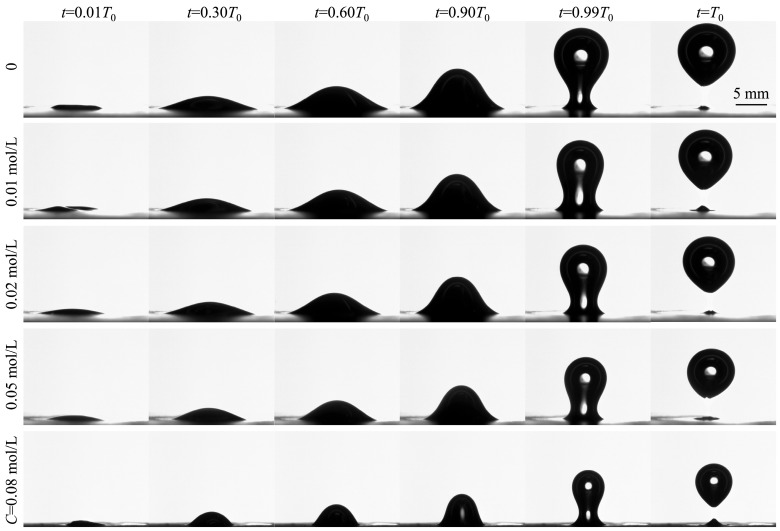
The time evolution of bubble shapes for five different 1-pentanol concentrations ranging from 0 to 0.08 mol/L. Time *t* = 0 is defined as the time of detachment of the previous bubble, and *T*_0_ is the bubbling period.

**Figure 4 biomimetics-10-00382-f004:**
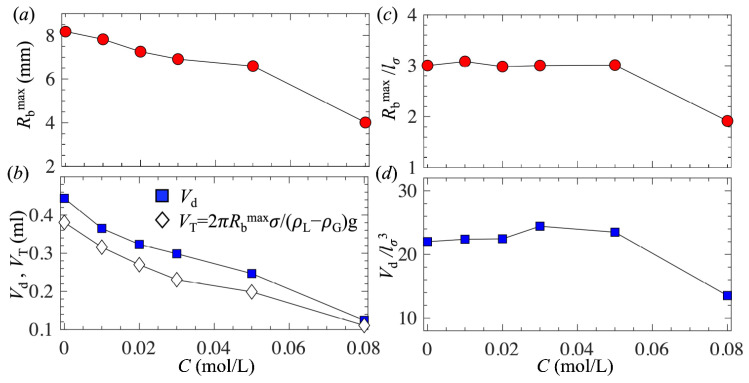
(**a**,**c**) The maximum bubble base radius as a function of the 1-pentanol concentration, and (**b**,**d**) the detached bubble volume as a function of the 1-pentanol concentration. In (**c**) and (**d**), *R*_b_^max^ and *V*_d_ are normalized by *l*_*σ*_ and *l*_*σ*_^3^, respectively, where *l*_*σ*_ = (*σ*/*ρ*_L_*g*)^0.5^ is the capillary length.

**Figure 5 biomimetics-10-00382-f005:**
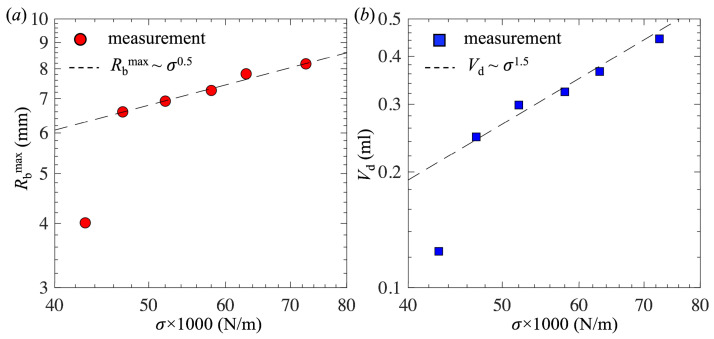
(**a**) The bubble maximum base radius and (**b**) bubble detached volume as a function of surface tension. Both axes are shown in log scale. The dashed lines are fitted to the experimental data in the large surface tension range.

**Figure 6 biomimetics-10-00382-f006:**
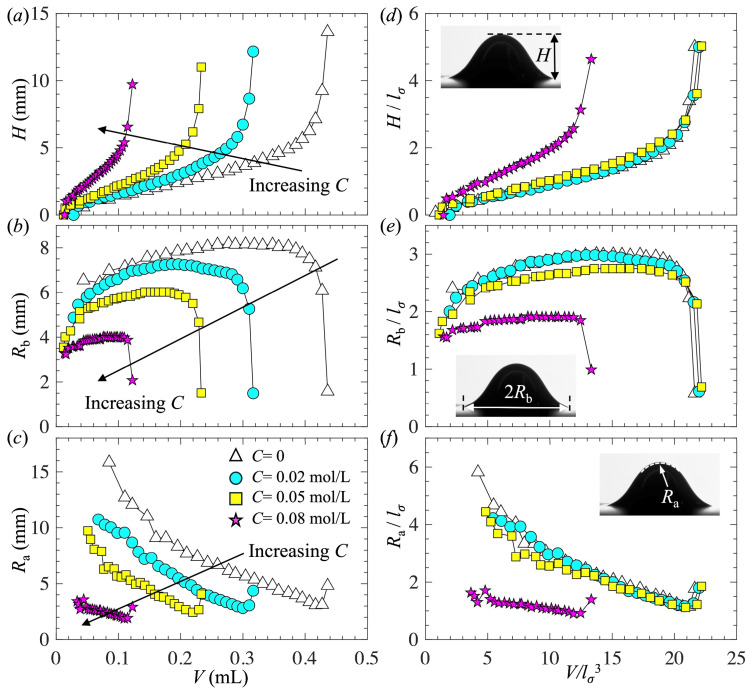
(**a**,**d**) Bubble height, (**b**,**e**) bubble base radius, and (**c**,**f**) bubble radius at the apex as a function of bubble volume. In (**d**–**f**), the length and volume are normalized by *l*_*σ*_ and *l*_*σ*_^3^, respectively.

**Figure 7 biomimetics-10-00382-f007:**
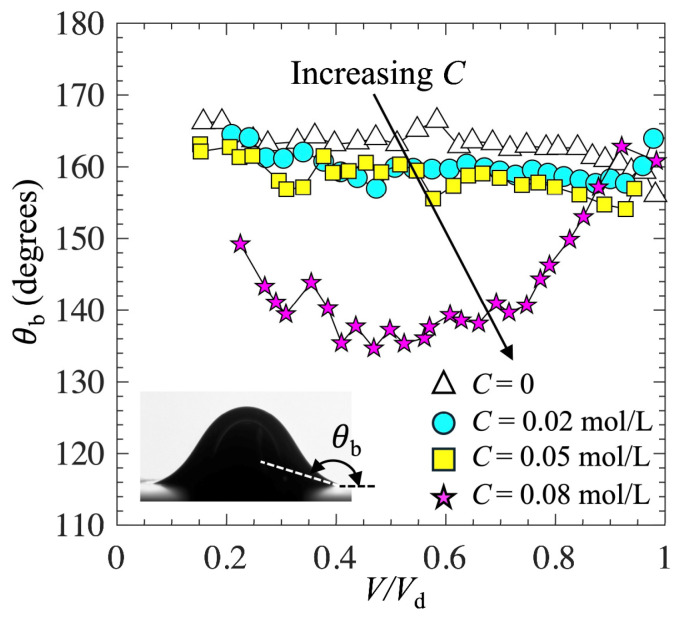
Evolution of contact angle at bubble base as function of bubble volume.

**Figure 8 biomimetics-10-00382-f008:**
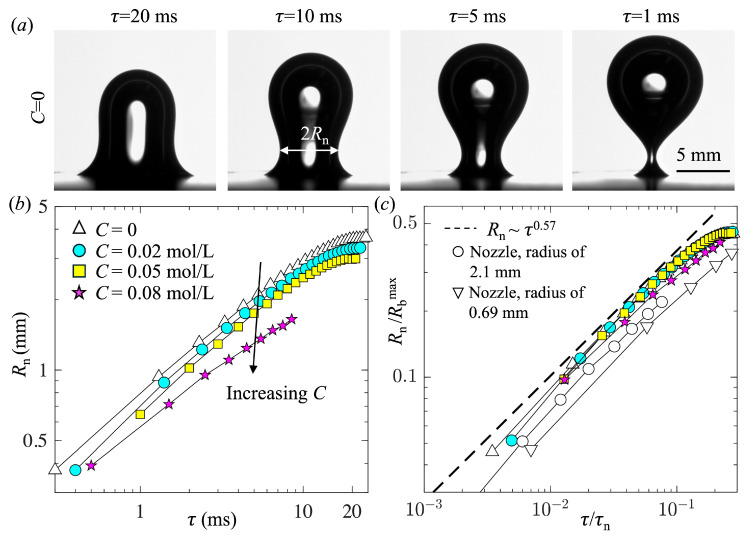
(**a**) The evolution of the bubble shape during the necking period for *C* = 0 and (**b**,**c**) variations in the minimum neck radius as a function of time to pinch-off. In (**c**), the length and time are normalized by *R*_b_^max^ and the capillary–inertia time scale, respectively. The results for bubble pinch-off from a nozzle obtained by Thoroddsen et al. [65] are also shown in (**c**). The dashed line shows the best power-law fitting the current data.

**Figure 9 biomimetics-10-00382-f009:**
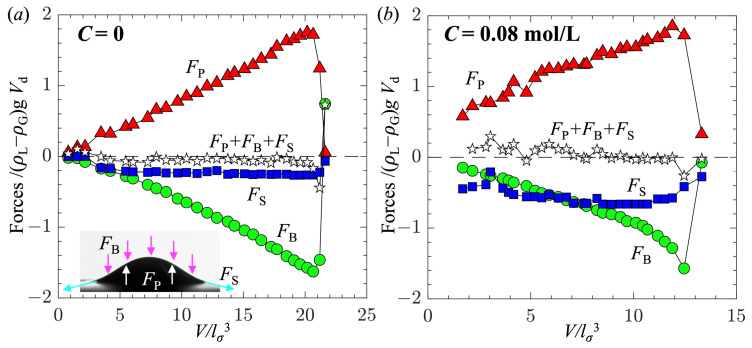
Variations in forces, including the pressure force *F*_P_, buoyancy force *F*_B_, surface tension force *F*_S_, and their summation *F*_P_ + *F*_B_ + *F*_S_, for (**a**) *C* = 0 and (**b**) *C* = 0.08 mol/L, as a function of bubble volume. The forces are normalized by (*ρ*_L_ − *ρ*_G_)*gV*_d_ and the volume is normalized by *l*_*σ*_^3^. The inserted image in (a) illustrated the forces applied on the bubble.

**Table 1 biomimetics-10-00382-t001:** The main experimental parameters and experimental results in the current study.

*C* (mol/L)	*σ* (mN/m)	*θ*_0_ (°)	*R*_b_^max^ (mm)	*V*_d_ (mL)	*l*_*σ*_ (mm)
0	72	159	8.2	0.44	2.7
0.01	63	159	7.8	0.37	2.5
0.02	58	158	7.3	0.32	2.4
0.03	52	156	6.9	0.30	2.3
0.05	47	155	6.6	0.25	2.2
0.08	43	131	4.0	0.12	2.1

## Data Availability

The original contributions presented in the study are included in the article, further inquiries can be directed to the corresponding authors.

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
