# Peer review of "Effect of Surfactant on Bubble Formation on Superhydrophobic Surface in Quasi-Static Regime"

_biomimetics, 2025, doi:10.3390/biomimetics10060382_

Round 1

Reviewer 1 Report

Comments and Suggestions for Authors

Influence of Nozzle Size:

The manuscript discusses bubble formation through a fixed 0.5 mm orifice but does not consider how varying nozzle sizes might affect the observed scaling laws for bubble detachment volume and base radius. How might the trends reported in Heidarian et al. (2025), 'Effects of nozzle size and surface hydrophobicity on microbubble generation' (https://doi.org/10.1063/5.0251293), impact the findings? Could the scaling laws proposed here be generalized for different nozzle sizes?

Additionally, the current study focuses solely on a single orifice size. Could the authors provide a rationale for selecting 0.5 mm as the nozzle size, and discuss whether the trends observed at this scale would hold for significantly smaller or larger orifices?

Hydrophobicity and Surfactant Interactions:

The study focuses on a single type of surfactant (1-pentanol) and its effect on SHS. Would the scaling laws hold if a more hydrophobic surfactant were introduced? How would the detachment volume and base radius change if a surfactant with a significantly lower surface tension than 1-pentanol was employed?

Furthermore, how would the results differ if the SHS surface were treated with a coating to further increase its hydrophobicity? Could the authors expand on how surface modifications, as discussed in Heidarian et al., might alter the observed scaling laws?

Pinch-Off Dynamics:

The manuscript briefly touches on the necking and pinch-off dynamics but does not quantitatively compare the neck radius evolution with established scaling laws in bubble dynamics literature. Could the authors include a more rigorous analysis of the necking process and compare it with the scaling relations observed in previous studies on nozzle-driven bubble formation?

Surface Tension Scaling Laws:

The manuscript presents the scaling laws as Rdmaxσ1/2 and Vdσ3/2. While these relations are theoretically sound, could the authors discuss potential deviations observed at higher surfactant concentrations? Is the reduced surface tension at C = 0.08 mol/L the only contributing factor, or are there other influencing parameters, such as changes in contact angle hysteresis or Marangoni effects, that could be considered?

Experimental Setup and Data Analysis:

The authors state that the equilibrium surface tension was determined using the pendant droplet method. Could more information be provided regarding the calibration and validation of this method? Were multiple trials conducted to confirm the consistency of the measured values, especially at lower surfactant concentrations?

Additionally, the study employs high-speed imaging to capture bubble dynamics. Could the authors provide more details on how image processing techniques were used to ensure accurate determination of geometrical parameters such as base radius and neck radius?

Reviewer 2 Report

Comments and Suggestions for Authors

This paper explores how the surfactant 1-pentanol affects bubble formation on superhydrophobic surfaces (SHS) under quasi - static conditions. Bubbles are formed by injecting gas through a nozzle on the SHS into a 1-pentanol aqueous solution at a constant flow rate. The focus is on bubble geometric parameters (size, contact angle, etc.) and force variations, aiming to reveal bubble formation mechanisms on complex surfaces in complex liquids. It is found that as surfactant concentration rises, bubble geometric parameters decrease. At low concentrations, the static contact angle θ₀ is nearly constant, bubble shapes are self - similar across concentrations, and bubble size is proportional to the capillary length. The research is distinctive, with rich experimental results. Before proceeding to publish, the following should be refined:

  1. Do different SHS texture geometries impact bubble formation? The authors should discuss the roles of surface morphology and roughness.
  2. Can quasi - static bubble formation results be generalized to dynamic conditions?
  3. Besides maintaining quasi - static conditions, the authors should detail how other variables (environmental temperature, solution purity, etc.) were controlled.
  4. Are the observed contact angle changes linked to surfactant adsorption on SHS?
  5. Based on experimental results, explore the mechanisms of surfactant effects on bubble formation, such as adsorption at the gas - liquid interface and impact on surface energy.
  6. Bubble adhesion and control on SHS is a key topic. Recent relevant literature is missing in the literature review(Advanced Functional Materials, 2019, 29(40): 1904766.Acs Applied Materials & Interfaces, 2021, 13(40): 48270-48280.ACS applied materials & interfaces, 2018, 10(19): 16867-16873.). The authors should compare with it to highlight their work's importance.

Round 2

Reviewer 1 Report

Comments and Suggestions for Authors

The manuscript significantly improved, and I think it can be published.